# RMAAT: A Bio-Inspired Approach for Efficient Long-Context Sequence Processing in Transformers

## Abstract

Astrocytes, an essential component of the brain's neural circuitry, demonstrate learning capabilities through bioplausible mechanisms such as presynaptic plasticity and hebbian plasticity. However, their integration into computational models remains underexplored. This paper advances astromorphic computing techniques to emulate transformer self-attention mechanisms, leveraging astrocytic nonlinearity and memory retention to improve long-range dependency processing in machine learning and natural language processing (NLP) tasks. Existing transformer models have difficulty handling lengthy contexts with thousands of tokens, even with substantial computational resources. We propose Recurrent Memory Augmented Astromorphic Transformers (RMAAT), integrating astrocytic memory and recurrent processing into self-attention, enabling longer context handling without quadratic complexity growth. Our bioplausible model has been found to outperform traditional transformers in experimental tests conducted on the Long Range Arena benchmark and IMDB dataset. Specifically, our model achieves a significant reduction in memory utilization and computational latency. This paves the way for biologically inspired AI models by illustrating how astrocytic characteristics may enhance the performance and efficiency of computational models.

## 1 Introduction

The human brain, a marvel of biological engineering, relies on a network of diverse cell types to perform complex functionalities. Among these cells, astrocytes are ubiquitous in critical regions essential for cognitive functions. Astrocytes, which are star-shaped glial cells, have long been thought to support neurons, but recent research has revealed that they also play an active role in various brain processes. They control synaptic activity, assist in learning and memory, and alter how information is processed and stored in the brain (Gibbs et al., 2008; Bohmbach et al., 2022; Oliveira & Araque, 2022). The diverse range of functions performed by astrocytes presents intriguing opportunities for incorporating their roles into computational models, providing novel pathways for enhancing artificial intelligence and machine learning (Kozachkov & Michmizos, 2017; 2020; González et al., 2012; Porto-Pazos et al., 2011).

Astromorphic computing aims to improve the efficiency and effectiveness of machine learning models by emulating the nonlinear processing and memory retention capabilities of astrocytes, (Mia et al., 2023; Kozachkov et al., 2023a). Integrating the principles of astromorphic computing with transformer architectures presents a novel approach to advancing artificial intelligence. Transformers have significantly impacted the domain of machine learning, specifically in natural language processing (NLP) (Vaswani et al., 2017). The self-attention mechanism is the key innovation of transformers; it allows models to determine the importance of each word in a sentence independent of their position. However, despite the remarkable success of transformers, challenges persist in efficiently handling long sequences. Conventional neural networks, including transformers, often struggle with lengthy data sequences due to architectural limitations (Wu et al., 2020; Bulatov et al., 2022). Similarly, brain-inspired models like spiking neural networks (SNNs) have advanced by emulating neuronal spiking behaviors and incorporating various bio-plausible synaptic learning rules—such as spike-timing-dependent plasticity, e-prop, DECOLLE, equilibrium propagation, among others (Lu & Sengupta, 2024; Bellec et al., 2020; Kaiser et al., 2020; Bal & Sengupta,

2022)—but they generally fail to fully integrate the essential role of astrocytes and the complex interactions within the tripartite synapse. These models frequently overlook astrocytes' capacity for recurrent processing and memory retention, as well as their function as temporal and spatial integrators (Kozachkov et al., 2023a; Mia et al., 2023; Han et al., 2023; Lines, 2025).

In this work, we leverage the memory retention and recurrent processing capabilities of astrocytes, macro-modeled through short- and long-term processes in the neuron-astrocyte network, to temporally integrate astrocytic functions within transformer architectures. We explicitly map astrocytic short-term plasticity to spatial relative information, an aspect previously overlooked. This mapping of short-term processes facilitates the integration of long-term plasticity, which is then incorporated to form temporal astrocytic memory, enabling the model to process long sequences by compressing context into memory tokens. We implement this compression mechanism derived from the neuroscience macro-model as a memory retention factor. Furthermore, leveraging astrocytic memory, we introduce the Astrocytic Memory Replay Backpropagation (AMRB) algorithm, which significantly reduces hardware computational burden. Through this neuroscience-algorithm co-design, we propose the Recurrent Memory Augmented Astromorphic Transformer (RMAAT), which offers a unique learning approach, achieving higher accuracy in long-context machine learning tasks with a fivefold reduction in hardware memory utilization.

## 2 Related Works and Main Contributions

While transformers have transformed sequence modeling with self-attention, their quadratic complexity limits scalability for long sequences (Vaswani et al., 2017). Efficient variants like Reformer (Kitaev et al., 2020) and Longformer (Beltagy et al., 2020) address this but often compromise model capacity and long-term dependency capture. Spiking Neural Networks (SNNs) offer energy-efficient alternatives (Pfeiffer & Pfeil, 2018), with advanced learning rules like e-prop (Bellec et al., 2020), DECOLLE (Kaiser et al., 2020), and surrogate gradients (Neftci et al., 2019). However, SNNs typically lack the capacity to handle long-term dependencies without major architectural changes and do not incorporate astrocytic functions or tripartite synapse interactions.

Research on integrating astrocytes into neural networks has predominantly focused on small-scale models for specific functions such as synaptic modulation and homeostasis. For example, Nadkarni & Jung (2007) explored astrocyte-mediated synaptic transmission, and Postnov et al. (2009) investigated astrocytic influence on neuronal synchronization. In addition, astrocytic models have been instrumental in advancing neuromorphic systems by enabling self-repair and improving robotic locomotion through homeostatic control of Central Pattern Generators (CPG). In self-repair mechanisms, astrocytes play a crucial role in restoring synaptic functions, as explored by (Wade et al., 2012) and (Han et al., 2023), where astrocytic retrograde signaling is used to repair damaged neural pathways. In CPG systems, astrocytes regulate neuronal synchronization and plasticity, facilitating gait generation and learning in robotic systems, as shown by (Han & Sengupta, 2023). Some other works in astromorphic computing have begun to explore astrocytic functions within machine learning models. Kozachkov et al. (2023a) introduced astrocyte-inspired neural networks focusing on tripartite synaptic plasticity, while Mia et al. (2023) leveraged astrocytic calcium dynamics to introduce nonlinearity into the self-attention mechanism of transformers, allowing the processing of information based on relative input positioning. However, they do not explicitly associate short-term plasticity with self-attention. Consequently, existing approaches tend to address isolated aspects of astrocytic function and fail to explicitly map both short-term and long-term plasticity onto machine learning models or integrate the temporal astrocytic plasticity necessary for efficient handling of long sequences.

Our work introduces a novel RMAAT architecture that leverages astrocytic plasticity for efficient processing of long-context sequences. The main contributions of this work are:

- **Temporal Memory via Long-Term Astrocytic Plasticity**: By integrating long-term astrocytic plasticity, we form temporal memory that retains context during sequential processing, enabling the model to handle long sequences by retaining contexts into memory tokens.

- **Innovative Astrocytic Memory Compression Mechanism**: We introduce a novel astrocytic compression mechanism that efficiently condenses contextual information into residual temporal memories.
- **Hardware Efficiency via Astrocytic Memory Replay Backpropagation (AMRB)**: By employing the AMRB learning technique, the proposed model achieves a $5\times$ reduction in hardware memory utilization compared to conventional transformers, enhancing practicality for resource-constrained applications.

## 3 METHODOLOGY

In order to introduce the Recurrent Memory Augmented Astromorphic Transformer (RMAAT), we first discuss the computational neuroscience backbone that forms the foundation of our model. This backbone creates the macro-model responsible for translating short and long-term synaptic plasticities into the astromorphic self-attention mechanism. We then leverage these plasticities within a machine learning framework to efficiently handle long-range dependencies, forming the RMAAT architecture that combines biologically-inspired mechanisms with transformer-based processing.

### 3.1 SHORT-TERM AND LONG-TERM PLASTICITY

To capture the spatial and temporal aspects of astrocytic plasticities, we focus on short-term plasticity (STP) and long-term plasticity (LTP). STP involves rapid synaptic adjustments, while LTP refers to lasting changes over hours to days, both critical for memory and learning (Min et al., 2012; Perez-Catalan et al., 2021; Pittà et al., 2015; Li, 2024; Alberini et al., 2018; Gordleeva et al., 2023; 2021). These plasticities integrate neuronal, synaptic, and astrocytic dynamics across different temporal and spatial scales (Kozachkov et al., 2023b; Kozachkov & Michmizos, 2017).

**Neuron-Astrocyte Network Dynamics:** The neuron-astrocyte network integrates the dynamics of neurons, synapses, and astrocytic processes shaped by short and long-term plasticities. Astrocytes modulate synaptic transmission and plasticity via gliotransmitter release and ionic concentration, stabilizing neuronal activity and forming attractor states—stable brain activity configurations encoding memories and behaviors (Becker et al., 2022; Mongillo et al., 2008). Using short-term plasticity, the network quickly adjusts to new information. These neuron-astrocyte interactions are modeled through differential equations, capturing the temporal evolution of synaptic weights and astrocytic influence (Kozachkov et al., 2023b).

**Short-term Process Dynamics:** We begin by examining the short-term processes that regulate synaptic weights via interactions between neurons, synapses, and astrocytes.

*(i) Neural Dynamics:* In the context of a spiking neuron model, the neuron-astrocyte network can be described by discrete events of neuron spikes and continuous synaptic interactions influenced by astrocytes. The membrane potential $V_i(t)$ of neuron $i$ evolves according to the following Eqn. 1:

$$\tau_n \frac{dV_i(t)}{dt} = -\lambda(V_i(t) - V_{\text{reset}}) + I_i(t) \tag{1}$$

Here, $\tau_n = R_m C_m$ is the neural dynamics timescale, which is the product of the membrane resistance $R_m$ and the membrane capacitance $C_m$. This timescale is in the order of short-term effects. $\lambda$ is the decay rate for the membrane potential. $V_{\text{reset}}$ is the reset potential, and $I_i(t)$ is the input current to neuron $i$ (Diehl et al., 2015). The input current $I_i(t)$ is determined by the synaptic inputs and an intrinsic bias as noted in Eqn. 2:

$$I_i(t) = \sum_{j=1}^{N} g(s_{ij}) S_j(t) + b_i, \quad S_j(t) = \sum_k \delta(t - t_k) \tag{2}$$

where, $S_j(t)$ represents the spike train from neuron $j$, defined as a sum of Dirac delta functions at spike times $t_k$. The synaptic weight $g(s_{ij})$ and the input bias $b_i$ influence the membrane potential dynamics. When the membrane potential $V_i(t)$ reaches a threshold $V_{\text{th}}$, the neuron fires a spike and the potential is reset to a reset potential $V_{\text{reset}}$. The activity of neuron $i$, denoted as $x_i$, is a function of its membrane potential $V_i$. Specifically, $x_i$ represents the firing rate or the probability of the neuron firing a spike at a given time, which is influenced by the membrane potential $V_i$.

*(ii) Synaptic Dynamics:* The variable $s_{ij}$ represents the level of synaptic facilitation, indicating the extent to which presynaptic spiking activity influences the postsynaptic neuron. Synaptic strength can either increase or decrease based on pre- and postsynaptic activity. Astrocytes modulate these synaptic interactions through mechanisms such as gliotransmitter release and ionic concentration changes, which in turn affect the synaptic weights $g(s_{ij})$. In accordance with studies on neuron-glial interactions (Kozachkov et al., 2023b; Gong et al., 2024), we consider tripartite synapses, where synaptic plasticity is regulated by an associated short-term astrocytic parameter, $p_{ij}^s$. The dynamics of synaptic facilitation are given by:

$$\tau_s \frac{ds_{ij}}{dt} = -\beta s_{ij} + \theta(x_i)\theta(x_j) + \psi(p_{ij}^s) + c_{ij} \tag{3}$$

In this equation, $\tau_s$ is the synaptic dynamics timescale. $\beta$ represents the decay rate of synaptic facilitation. The function $\theta(x)$ captures the nonlinear interactions between these biological variables. The variables $x_i$ and $x_j$ represent the activity levels of neurons $i$ and $j$, respectively. The constant $c_{ij}$ acts as a bias, setting the baseline rate of synaptic facilitation in the absence of external input. $p_{ij}^s$ can be understood as the concentration of intracellular $Ca^{2+}$ ions in the astrocytic process surrounding the synapse. Astrocytes modulate neural activity through $Ca^{2+}$-dependent release of gliotransmitters such as GABA, D-serine, ATP, and glutamate (de Ceglia et al., 2023)—this modulation is captured by the function $\psi(p_{ij}^s)$.

*(iii) Short-Term Astrocytic Process Dynamics:* The state of a particular astrocytic process is influenced by its interactions with neurons at the tripartite synapse and with other astrocytic processes via intracellular calcium transport. This relationship is described by the equation:

$$\tau_p^s \frac{dp_{ij}^s}{dt} = -\gamma^s p_{ij}^s + \sum_{k,l=1}^{N} T_{ijkl}\psi(p_{kl}^s) + d_{ij} \tag{4}$$

Here, $\tau_p^s$ represents the timescale for short-term astrocyte dynamics, and $\gamma^s > 0$ is a decay term for the intracellular calcium in the astrocytic process. The double sum captures the interactions between the process $p_{ij}^s$ and all other processes. In the simplest case, calcium can diffuse between processes, represented by a non-linear function $\psi(p)$ and the tensor $T_{ijkl}$ describing concentration fluxes. The term $T_{ijkl}$ is a parameter that captures the spatial positions for the multiple neurons connected to that tripartite synapse. and encodes the spatial distance between the processes $p_{ij}^s$ and $p_{kl}^s$.

While previous efforts have implicitly incorporated short-term process dynamics, we are the first to explicitly propose and map these processes to a neuron-astrocyte network, which subsequently enable us to integrate astrocytic temporal dynamics into the proposed RMAAT architecture. Next, we address the *long-term process dynamics*, a key contribution that enables the temporal integration of astrocytes within the RMAAT architecture.

**Long-Term Process Dynamics:** The state of a particular astrocytic process for long-term dynamics is described by the equation:

$$\tau_p^l \frac{dp_{ij}^l}{dt} = -\gamma^l p_{ij}^l + \kappa(s_{ij}) \tag{5}$$

Here, $\tau_p^l$ represents the timescale for long-term astrocyte dynamics and is greater than the timescale for short-term astrocyte dynamics. $\gamma^l > 0$ is a decay term for the intracellular calcium in the astrocytic process. The nonlinear function $\kappa(s_{ij})$ represents the synapse-to-astrocyte signaling pathway at the tripartite synapse level, capturing the effects of synaptic activity on astrocytic processes.

In long-term dynamics, the astrocyte process is performing a form of temporal integration, where it accumulates and integrates the effects of sustained synaptic activity over time. This is essential for the persistent changes associated with long-term plasticity (LTP) and is driven by complex biochemical pathways influenced by the history of synaptic activity. This temporal integration is supported by research showing that long-term changes in astrocytic function involve prolonged biochemical signaling and gene expression changes, which are critical for maintaining long-term synaptic plasticity (Nedergaard et al., 2003; Perea et al., 2009).

**Synergy of Short-Term and Long-Term Dynamics:** We developed a macro-model to simulate the interaction between short-term plasticity (STP) and long-term plasticity (LTP) using a multi-neuron astrocyte network, consisting of 3 presynaptic ($V_j$) and 3 postsynaptic neurons ($V_i$) connected by

9 synapses ($s_{ij}$). The network includes a single astrocyte governing 9 processes in both short-term ($p_{ij}^s$) and long-term dynamics ($p_{ij}^l$). We map these process parameters as attractor states in this macro-model. Neural ($V_{i,j}$), synaptic ($s_{ij}$), and short-term astrocytic processes ($p_{ij}^s$) are modeled with the STP timescale, while long-term astrocytic processes ($p_{ij}^l$) follow the LTP timescale. The details of hyperparameters describing the timescale, membrane potential threshold, etc. are discussed in Appendix C.

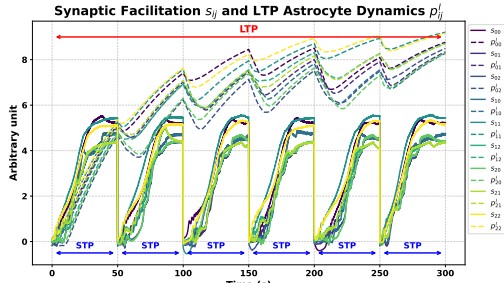

One STP cycle is defined as complete when the attractor states of the network stabilize, which, in this model, takes 50 seconds. The dynamics, described by Eqns. 1 - 5, were simulated for 300 seconds, encapsulating 6 STP cycles within one LTP cycle. The long-term astrocytic process parameter $p_{ij}^l$ reaches a stable attractor state at the 300s mark, accumulating information from the short-term parameters as calcium ions build up during each STP cycle. Depending on the LTP timescale ($\tau_p^l$), a varying number of STP cycles can fit within one LTP cycle. We later map the characteristics of STP and LTP in RMAAT, where each STP represents a segmented forward pass within a sequence involving long-range context. To conceptualize segmented processing, the current STP cycle is reset upon reaching a stable attractor state, after which a new STP cycle is initiated. As depicted in Fig. 1, after every 50 seconds, a new STP cycle begins, with $p_{ij}^l$ continuing to accumulate information from short-term parameters ($s_{ij}, p_{ij}^s$). As $p_{ij}^s$ and $s_{ij}$ show similar characteristics in STP, for simplicity, only $s_{ij}$ and $p_{ij}^l$ are depicted in Fig. 1. Although arbitrary units have been used to plot the parameters, $s_{ij}$ and $p_{ij}^l$ can be mapped to synaptic weights and calcium concentration respectively.

Figure 1: Simulated dynamics of synaptic facilitation $s_{ij}$ (STP) and astrocyte process $p_{ij}^l$ (LTP) in the macro-model. The figure highlights the rapid stabilization of STP, while LTP gradually builds up over multiple cycles, ultimately reaching a stable attractor state, with cumulative STP effects contributing to the LTP process.

## 3.2 Neuron-Astrocyte Network

To utilize STP and LTP processes of the neuroscience macro-model in machine learning, a neuroscience-algorithm co-design framework must be established, and neuron-astrocyte networks offer a viable approach for this integration. Recent works on neuron-astrocyte networks have proposed decomposing the softmax attention into linearized kernels by drawing inspiration from biologically plausible plasticities and nonlinearities observed in astrocytes (Kozachkov et al., 2023a; Mia et al., 2023). This approach decomposes the self-attention formulation into a kernelized operation

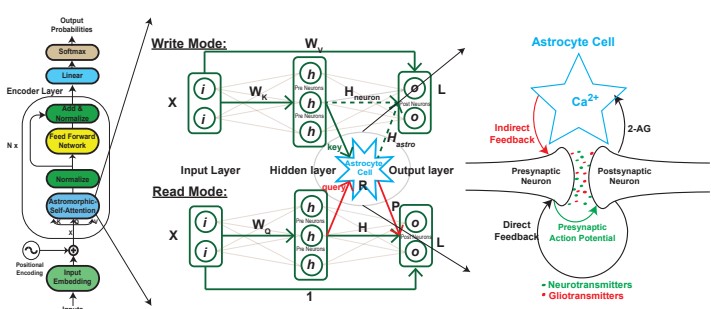

Figure 2: Overview of the Astromorphic Transformer architecture. This diagram illustrates the integration of bioplausible bi-directional feedback mechanisms within a two-layered neuron-astrocyte network, emulating the self-attention of the transformer encoder. The tripartite synapse between the hidden and output layers is shown, with a detailed depiction of the biological tripartite synapse on the right. The synaptic weights $W_K$, $W_Q$ and $W_V$ are corresponding to $K$, $Q$ and $V$ of the transformer. $2 - AG$ initiates the bidirectional communication between astrocyte and neuron which is linked to the direct and indirect feedback evoked by the $Ca^{2+}$ concentration.

in transformers. The decomposition is carried out by a two-layer neuron-astrocyte network that operates in both write and read modes. In the write mode, information is encoded as keys and

values, while in the read mode, it is decoded using queries within the transformer architecture. The depiction in Fig. 2 illustrates how bioplausible plasticities influence the network during both the write and read modes.

In the tripartite synapse of Fig. 2, the presynaptic action potential activates both the postsynaptic neuron and the astrocyte, triggering a calcium response in the astrocyte. This results in direct feedback from the postsynaptic neuron and indirect feedback mediated by the astrocyte. The dynamics of neuron-astrocyte communication can be referenced from Li and Rinzel $Ca^{2+}$ dynamics (Li & Rinzel, 1994), gatekeeper model (Volman et al., 2007), Nadkarni and Jung model (Nadkarni & Jung, 2004; 2007), and bidirectional study on astrocytic glutamate binding to postsynaptic neurons (AN model) (Wade et al., 2011). The comprehensive description of the computational model can be found in the works of (Wade et al., 2012; 2011; De Pittà et al., 2009; Navarrete & Araque, 2010).

In the neuron-astrocyte network of Fig. 2, the write mode encodes input sequences by adjusting synaptic weights through biologically inspired plasticity mechanisms. During the write mode, the input for token $t$ is processed in the neuron-astrocyte network (Fig. 2) as follows: $i_t = x_t$, $h_t = \phi(i_t W_K) = \phi(k_t)$, $o_t = i_t W_V = v_t$. During the write mode, the input layer sequentially transmits the keys $k_t = i_t W_K$ to the hidden layer and the values $v_t = i_t W_V$ to the output layer. The hidden layer neurons activate a non-linear feature map $\phi(.)$ on the keys to create outputs $h_t = \phi(k_t)$, and the values $v_t$ are transmitted to the output layer. $\phi(.)$ can be implemented using radial basis function (RBF) (Peng et al., 2021), rectified, or exponent-based (Katharopoulos et al., 2020) kernels. We employ the feature map $\phi(x) = \text{elu}(x) + 1$ due to its superior performance compared to alternative kernels.

In the following subsections, we explain in detail how these astrocytic plasticities, specifically hebbian and presynaptic plasticites, are incorporated during the write mode and their role in enabling the self-attention mechanism in the read mode. Moreover, we emphasize our contribution in mapping short-term processes through the spatial integration of astrocytes within the neuron-astrocyte network, culminating in the development of the RMAAT architecture. To map STP into astromorphic computing, we first outline the specific plasticity mechanisms associated with the neuron-astrocyte network.

**Hebbian Plasticity:** Hebbian plasticity adjusts synaptic weights based on the co-activation of pre and postsynaptic neurons along with astrocytes. Two forms of Hebbian plasticity are modeled: *(i) Neuronal Hebbian Plasticity:* The synaptic weight between these neurons, $H_{neuron}$, is updated based on the co-activation of these neuron layers: $H_{neuron,t} = H_{neuron,t-1} + \frac{1}{m} h_t^T o_t$. where, $m$ represents the number of neurons in the hidden layer for each token. We map this $H_{neuron,t}$ to $\theta(x_i) \cdot \theta(x_j)$ in the synaptic dynamics of our proposed macro-model (Eqn. 3), representing the interaction between the activations of neurons $i$ and $j$, capturing the essence of hebbian learning where the connection strength increases as the neurons co-activate. *(ii) Astrocytic Hebbian Plasticity:* Astrocytes modulate synaptic efficacy through gliotransmitter release, which is modeled by the weight $H_{astro}$. This weight is updated as follows: $H_{astro,t} = H_{astro,t-1} + \frac{1}{m} \phi(R)^T o_t$, where $R$ represents the astrocytic activity, influenced by calcium dynamics. $\phi(R)$ reflects the indirect feedback from astrocytes to postsynaptic neurons. We propose to associate $H_{astro}$ with the term $\psi(p_{ij}^s)$ in the synaptic dynamics of the neuroscience macro-model, where the activation of $p_{ij}^s$ ($\psi(p_{ij}^s)$) processes the inherent information of $T_{ijkl}$.

**Presynaptic Plasticity:** Presynaptic plasticity is influenced by the presynaptic activity (neurotransmitter release) and the astrocytic calcium dynamics, which regulates the release of neurotransmitters. The presynaptic plasticity parameter $g_t$ is the link between these two entities that captures the concentration of neurotransmitter in the synaptic cleft based on the neuron firing activity, $x_i$ ($x_i$ is modulated by membrane potential, $V_i$ in Eqn. 1). Thus, $g_t$ accumulates the presynaptic activity as input tokens are processed: $g_t = g_{t-1} + h_t = g_{t-1} + \phi(k_t)$. The non-linear response of astrocytes to presynaptic activity is modeled by raising this sum to the power of $\alpha$, a hyperparameter that encodes the non-linearity of calcium dynamics (Mia et al., 2023): $g = \left( \sum_{t=1}^{N} \phi(k_t) \right)^{\alpha}$. This non-linear presynaptic plasticity reflects the slowing of calcium accumulation as presynaptic activity intensifies.

Both plasticities are encoded sequentially in the write mode through keys and values. However, in the read mode, the queries are presented in a parallel manner, where $Q = \begin{pmatrix} q_1 \\ q_2 \\ .. \\ q_N \end{pmatrix}$ is a matrix of shape

$R^{N \times d}$, with $N$ representing the number of tokens and $d$ as the embedding dimension for each token. During decoding, information is extracted in parallel, with $K$ and $V$ sharing the same dimensions as $Q$. This process ultimately enables the realization of astromorphic self-attention.

**Realization of Astromorphic Self-attention:** As discussed previously, after the Hebbian and presynaptic plasticities are encoded in the write mode, the read mode retrieves the stored information to compute self-attention. The hidden layer activates queries, $(h = \phi(Q))$ and the encoded information is accessed through the combined hebbian weight $H$ and the presynaptic plasticity parameter $g$. In Eqn. 6, $H$, which is the sum of both neuronal and astrocytic hebbian plasticities, captures the plasticities for all the $N$ tokens, thus is represented by matrix form.

$$H = H_{neuron} + H_{astro} = \frac{1}{m} \left( \phi(K)^T V + \phi(R)^T V \right) \tag{6}$$

Here, $\phi(R)$ captures the relative position information between tokens which we map to $T_{ijkl}$ in the neuroscience macro-model. The read mode computes the calcium response $C = hg^T$ and the weight $P$, modulated by astrocytic activity, is computed based on calcium response $P = \frac{1}{C}$ for $N$ tokens in Eqn. 7.

$$P = \frac{1}{\phi(Q) \left( \sum_{t=1}^{N} \phi(k_t) \right)^\alpha} \tag{7}$$

Finally, the output of the network during the read mode is computed by applying the learned hebbian weight $H$ and the presynaptic weight $P$ to the hidden-to-output layer connection, with a residual connection from the input layer: $L = \phi(Q)(H \odot P) + I$, where $\odot$ denotes element-wise multiplication, and $I = X$ represents the direct residual connection from the input layer. By integrating the Hebbian plasticity $H$ and presynaptic plasticity $P$, we illustrate the final output of the astromorphic self-attention mechanism in Eqn. 8:

$$L = \frac{\phi(Q)(\phi(K)^T V + \phi(R)^T V)}{m \times \phi(Q) \sum_{t=1}^{N} [\phi(k_t)]^\alpha} + X \tag{8}$$

To complete the transformer operation, layer normalization is applied to the self-attention output $L$, followed by a feedforward network (FFN) to produce the final logits in Eqn. 9.

$$Y = \text{LayerNorm}(L), \ \text{logits} = \text{Softmax}(\text{Linear}(\text{FFN}(Y) + Y)) \tag{9}$$

The final output logits represent the classification probabilities or other task-specific outputs, depending on the application of the astromorphic transformer.

**Spatial Mapping:** In the macro-model, $T_{ijkl}$ is responsible for spatial encoding, which is utilized through $R$ in $H_{astro}$. We capture this spatial mapping by simulating the macro-model for various spatial orientations of neurons in $T_{ijkl}$ and tracking the resulting outputs of $p_{ij}^s$. Different spatial configurations produce varying responses in $p_{ij}^s$ and neuron spiking activity. However, when spatial distances remain constant, $p_{ij}^s$ shows no variation, and neuron spiking activity remains uniform. So, we can deduce an impact of relative spatial position on the neuron-astrocyte network that is mapped by $T_{ijkl}$ to $H_{astro}$ and then to $R$. Consequently, we establish a previously unexplored connection between the neuroscience macro-model and relative positional encoding in astromorphic self-attention. Prior approaches did not explicitly investigate how short-term processes could be mapped to spatial information. By addressing this gap, we have enabled our model to exploit long-term processes for the integration of temporal information, achieving a more comprehensive representation in our current approach. In the following sections, we explore the temporal integration of astrocytes into astromorphic computing by introducing astrocytic memory as memory tokens. Additionally, we propose the RMAAT architecture, which effectively handles long-context sequences using astrocytic memories within a machine learning framework, incorporating an astrocytic compression algorithm and the AMRB learning rule.

### 3.3 Temporal Integration of Astrocytes in Transformers

Temporal integration in astrocytes originates from long-term tripartite synaptic plasticity, represented by the long-term astrocytic process parameter $p_{ij}^l$ in Eqn. 5. The timescale $\tau_p^l$, being significantly longer than $\tau_p^s$ in Eqn. 4, allows for extended retention of information. This accumulation of $Ca^{2+}$ responses constitutes the long-term astrocytic process, $p_{ij}^l$, which gives rise to the concept of astrocytic memories. To incorporate this feature into transformer architectures, we introduce the notion of memory tokens based on these astrocytic processes.

**Memory tokens:** Memory tokens are designed to persist through multiple processing stages of serialized sequences. Traditional memory-augmented transformer architectures initialize and learn these tokens externally, passing them from one segment to another in segmented sequence propagation (Bulatov et al., 2022; Wu et al., 2020). However, they often lack the context needed to establish a cohesive meaning. These architectures incur greater computational costs due to BPTT and external memory initialization. In contrast, our approach uses astrocytic memories as naturally integrated tokens within the neuron-astrocyte network's long-term processes, eliminating the need for external initialization and reducing computational overhead.

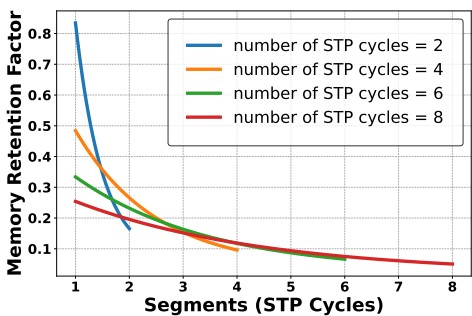

Figure 3: Memory retention factor of astrocytes demonstrating the compression mechanism to emulate long-term plasticity based on the macro-model. The graph illustrates how the retention factor decreases as the number of segments (STP cycles) increases, indicating that more segments lead to higher compression for each segment.

**Astrocytic Memory:** Our proposed neuroscience macro-model simulates the behavior of the synaptic facilitation $s_{ij}$ and the long-term astrocytic process parameter, $p_{ij}^l$, which represents the astrocytic memory accumulation over time. This macro-model accounts for all the neural, synaptic, and astrocytic dynamics described in Eqns. 1 - 5. Based on the behavior of $p_{ij}^l$ in Fig. 1, we observe that with repeated short-term plasticity (STP) events, the parameter gradually saturates, as calcium ions are finite, leading to a stabilized state in the calcium response. We model the behavior of $p_{ij}^l$ by interpolating the LTP response from Fig. 1, allowing us to map bioplausibility onto the astrocytic memory tokens.

Based on the interpolated LTP dynamics in Fig. 1, we propose an algorithm that compresses stored information by scaling the memory tokens. The compression is directly influenced by the number of STP cycles encoded within a single LTP cycle. A model is fitted to the interpolated LTP curve, where the area under the curve is normalized to 1, representing the total available $Ca^{2+}$ ion concentration. For each STP cycle, the $Ca^{2+}$ response constitutes a fraction of this total calcium, with the fraction decreasing as the number of STP cycles increases. If all STP fractions are integrated, they sum to the total $Ca^{2+}$ ion concentration of 1. This fraction is termed the memory retention factor, as it represents the $Ca^{2+}$ ion concentration corresponding to the long-term astrocytic process parameter, $p_{ij}^l$. This compressed astrocytic memory effectively manages long-range dependencies without requiring external memory initialization, reducing computational overhead and improving scalability. As the number of STP cycles increases, the memory retention factor decreases, resulting in a more compressed form of astrocytic memory, as shown in Fig. 3. The number of STP cycles corresponds to the number of segments used for a given machine learning dataset in RMAAT. The approach for dividing long input sequences into multiple segments, based on the RMAAT's model capacity, is explained in the next section.

### 3.4 THE RECURRENT MEMORY AUGMENTED ASTROMORPHIC TRANSFORMER (RMAAT)

The Recurrent Memory Augmented Astromorphic Transformer (RMAAT) is specifically engineered to effectively process lengthy sequences by partitioning them into smaller segments. The maximum sequence length for each segment is established according to the model's token handling capacity. These segments are then processed in a temporally recurring manner, beginning with the first segment. These segments map to the STP cycles in the neuroscience macro-model in Fig. 1. As the RMAAT processes each segment, it utilizes the astrocytic memory compression algorithm discussed above and propagates the compressed contexts in form of memory tokens, as depicted in Fig. 4.

**Temporal Recurrence and Context Propagation:** As subsequent segments are processed, the inherent compressed memory stored in the astrocytes facilitates the transition of context from one segment to the next. The recurrent nature of the processing guarantees that the context from prior segments is consistently transmitted, preserving a unified comprehension of the entire sequence. By employing this technique of connecting astrocytic memory tokens, the model is able to effortlessly include information from different segments, efficiently handling distant connections without sacrificing important contextual details. The time-unrolled diagram in Fig. 4 illustrates the progression

of compressed astrocytic memories throughout the sequence. The $mem_t$ tokens represent the compressed memories, following the compression algorithm outlined in Fig. 3, where $t$ denotes the identity of STP cycles, corresponding to the segments in RMAAT. Astrocytic memories also enable the development of an efficient learning algorithm by utilizing the stored memory tokens.

**Resource-Efficient Learning with AMRB:** The Recurrent Memory Augmented Astromorphic Transformer (RMAAT) employs an innovative approach to learning known as Astrocytic Memory Replay Backpropagation (AMRB), which draws inspiration from the learning algorithms used by recurrent neural networks (Wu et al., 2020; Wan et al., 2023). In AMRB, all memory tokens are saved and replayed backward during the gradient calculation phase (See Appendix A for the detailed algorithm). AMRB aligns closely with biological systems through its use of localized computation, where neurons and astrocytes interact locally to manage memory and learning. Unlike BPTT, which propagates information globally across all time steps, AMRB leverages localized astrocytic memories that store compressed information from previous interactions. This localized approach mirrors the way biological systems handle information, as astrocytes modulate synaptic activity based on local neural activity without retaining the entire sequence state. As a result, AMRB not only enhances bio-plausibility but also significantly reduces

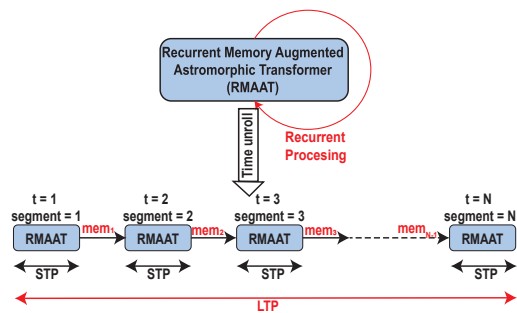

Figure 4: Illustration of the Recurrent Memory Augmented Astromorphic Transformer (RMAAT) processing through time unrolling. The diagram demonstrates how segments are processed sequentially with recurrent processing, where astrocytic memory ($mem_t$) flows from one segment to the next, preserving context throughout. Short-term plasticity (STP) modulates information within each segment, while long-term plasticity (LTP) supports the recurrent processing and memory flow.

computational overhead and memory requirements. This reduction in resource usage makes AMRB more energy-efficient, as it minimizes the power and memory consumption typically associated with BPTT, further optimizing transformer models for resource-constrained environments.

AMRB reduces hardware memory usage by a factor of five, significantly lowering the memory burden during the backward pass. Additionally, as the memory tokens are stored inherently in astrocytes rather than externally provided, AMRB improves computational efficiency, running approximately 15% faster than BPTT, which results in higher processing speeds without sacrificing accuracy.

## 4 RESULTS

In this section, we present the performance evaluation of RMAAT across two key benchmarks: the IMDB dataset for sentiment classification (Maas et al., 2011) and the Long Range Arena (LRA) for text classification (Tay et al., 2020). These benchmarks were chosen to assess the model's ability to handle tasks requiring both short-term and long-term context understanding. The detailed experimental setup is outlined in Appendix B. The models' performance was evaluated using key metrics: accuracy, memory usage, and computational speed. Accuracy reflects the proportion of correct predictions, indicating model performance. Memory utilization assesses hardware efficacy during training and inference, while computational speed evaluates data processing in relation to the Softmax Transformer baseline (1x). RMAAT's comparison with other transformers underscores its enhanced precision and resource efficiency, particularly for long-range assignments.

### 4.1 SENTIMENT CLASSIFICATION ON IMDB

To evaluate the performance of RMAAT on sentiment classification, we conducted experiments using the IMDB dataset. We used 256 as the maximum sequence length allowed for each RMAAT block depicted in Fig. 4. The results are summarized in Table 1, comparing the performance of various transformer models and memory-augmented variants (The hyperparameters are listed in Appendix C).

When processing a single segment, the astromorphic transformer demonstrates comparable performance to the Softmax and Linear Transformers. However, its true advantage emerges with longer sequences, where the RMAAT outperforms both the RMT and Linear Transformer variants. Additionally,

Table 1: IMDB Sentiment Classification Results

| Model | Seg-ments | Recur-rency | Accuracy (% ± std) | Memory (GB) | Speed |
|---|---|---|---|---|---|
| Softmax Transformer | 1 | No | 85.5 ± 0.1 | 6.1 | 1x |
| Astromorphic Transformer(Mia et al., 2023) | 1 | No | 85.8 ± 0.2 | 7.1 | 0.92x |
| Linear Transformer(Kozachkov et al., 2023a) | 1 | No | 85.7 ± 0.2 | 6.6 | 0.95x |
| Spike-Transformer(Mueller et al., 2021) | 1* | No | 86.36 | - | - |
| Recurrent SNN (Wang et al., 2022) | 1* | No | 86.82 | - | - |
| CoRNN (Rusch & Mishra, 2020) | 1* | No | 87.4 | - | - |
| LSTM (Aswani et al., 2021) | 1* | No | 87.48 | - | - |
| SNN (Agrawal et al., 2021) | 1* | No | 88.15 | - | - |
| RMT variant (Softmax) (Bulatov et al., 2022) | 6 | Yes | 88.4 ± 0.2 | 20.2 | 1x |
| Linear Transformer | 6 | Yes | 88.2 ± 0.2 | 22.5 | 1.03x |
| **RMAAT (Our Model)** | 6 | Yes | **88.9 ± 0.1** | **4.4** | **1.15x** |

*\* These models are not iso-architecture and may process longer sequence lengths*

RMAAT shows significant memory efficiency, achieving approximately a fivefold reduction in memory usage compared to other models while retaining superior contextual information. RMAAT also processes data 15% faster than the RMT variant and 10% faster than the Linear Transformer. Compared to models like Spike-Transformer, Recurrent SNN, and LSTM, RMAAT offers clear advantages in both accuracy and efficiency. These models have varying architectures compared to ours as they are taken from different literature.

## 4.2 Text Classification on Long Range Arena Benchmark

The Long Range Arena (LRA) benchmark evaluates models on their ability to handle long-range dependencies. The text classification task involves sequences much longer than typical benchmarks, making it ideal for assessing RMAAT's capacity to maintain context over extended lengths. Sequences were tokenized

Table 2: LRA Text Classification Results

| Model | Seg-ments | Recur-rency | Accuracy (% ± std) | Memory (GB) | Speed |
|---|---|---|---|---|---|
| Softmax Transformer (Mia et al., 2023) | 1 | No | 61.5 ± 0.1 | 6.3 | 1x |
| Astromorphic Transformer (Mia et al., 2023) | 1 | No | 61.5 ± 0.1 | 7.6 | 1.9x |
| Linear Transformer (Mia et al., 2023) | 1 | No | 61.0 ± 0.1 | 5.9 | 1.01x |
| Local Attention (Vaswani et al., 2017) | 1* | No | 52.98 | - | - |
| Sparse Transformer (Child et al., 2019) | 1* | No | 63.58 | - | - |
| Longformer (Beltagy et al., 2020) | 1* | No | 62.85 | - | - |
| Linformer (Wang et al., 2020) | 1* | No | 53.94 | - | - |
| Reformer (Kitaev et al., 2020) | 1* | No | 56.10 | - | - |
| Synthesizer (Tay et al., 2021) | 1* | No | 61.68 | - | - |
| BigBird (Zaheer et al., 2020) | 1* | No | 64.02 | - | - |
| Performer (Choromanski et al., 2020) | 1* | No | 65.40 | - | - |
| RMT variant (Softmax) (Bulatov et al., 2022) | 8 | Yes | 65.0 ± 0.2 | 24.0 | 1x |
| Linear Transformer | 8 | Yes | 64.8 ± 0.1 | 22.6 | 1.13x |
| **RMAAT (Our Model)** | 8 | Yes | **65.9 ± 0.1** | **5.1** | **1.5x** |

*\* These models are not iso-architecture and may process longer sequence lengths*

at the byte level with up to 512 tokens per segment, and a total of 4096 tokens (8 segments). All the hyperparameters are listed in Appendix C. Table 2 summarizes the results, comparing RMAAT with other transformer models and memory-augmented variants on the LRA text dataset. In the evaluation with 8 segments, RMAAT demonstrates superior performance, operating 50% faster than the RMT variant and 30% faster than the Linear Transformer, while also significantly reducing memory usage to 5.1 GB. Compared to other efficient transformers like Longformer, Linformer, and Sparse Transformer, RMAAT excels in handling long-range dependencies with enhanced computational efficiency. The inclusion of Astrocytic Memory Replay Backpropagation (AMRB) strengthens the model's ability to manage complex, long-context sequences effectively, showcasing the impact of bioplausible mechanisms in transformer architectures.

## 5 Conclusions

This research introduces the RMAAT, a transformer enhanced by bioplausible astrocytic mechanisms. By integrating short and long-term astrocytic plasticity, RMAAT addresses long-range dependencies in sequential data, outperforming standard transformers in both accuracy and efficiency. With AMRB, the model reduces memory usage and enhances computational speed, making it well-suited for resource-limited systems. Moving forward, further biological inspiration, such as astrocyte-astrocyte communication through gap junctions, can be explored as potential avenues to parallelize sequential processing and enhance computational efficiency. Future work can also extend RMAAT to more challenging domains like computer vision and machine translation, aiming to create more efficient, biologically inspired AI systems that closely emulate human cognition.

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

## A  ASTROCYTIC MEMORY REPLAY BACKPROPAGATION (AMRB)

This algorithm describes the Astrocytic Memory Replay Backpropagation (AMRB), which efficiently manages and updates memory states across multiple segments. By leveraging the bioplausible properties of astrocytes, AMRB ensures the preservation and propagation of context throughout the network, thereby enhancing the model's ability to handle long-range dependencies. The algorithm is outlined in Alg. 1.

## B  EXPERIMENT SETUP

**Hardware Specifications & Network Architecture:** All experiments were conducted on a server equipped with Linux 5.15.0-113-generic x86_64 with glibc2.31. The server featured an AMD EPYC 7742 64-Core Processor with 32 CPU cores, 1TB of RAM, and 8 NVIDIA RTX A5000 GPUs, each with 24GB of memory. The RMAAT network architecture, depicted in Fig. 4, can have varying numbers of encoder layers depending on the ML task. The model leverages the inherent memory and processing capabilities of astrocytes, with recurrent processing as depicted in Fig. 4. The specific hyperparameters, such as the number of layers, heads, and size of the hidden states, etc. are detailed in the supplementary section.

**Datasets:** The IMDB dataset consists of 50,000 movie reviews, split evenly into 25,000 training and 25,000 testing samples. Each review is labeled as either positive or negative, providing a balanced dataset for sentiment classification. This dataset is particularly useful for evaluating how well models can capture and retain context over relatively long sequences, as movie reviews often contain nuanced and varied language. The LRA benchmark includes several tasks designed to test the ability of models to handle long-range dependencies. For the text classification task, sequences are significantly longer than those in standard benchmarks, making it an ideal test for evaluating the RMAAT's capability to maintain context over extended lengths.

---

**Algorithm 1:** Astrocytic Memory Replay Backpropagation (AMRB)

---

**Input:** $rollout = [x_0, x_1, \ldots, x_T]$: A list containing each timestep $t$'s input $x_t$
**Input:** $prev\_memory$: Memory from the previous rollout
**Result:** Updated memory for the next rollout

1 Initialize $replay\_buffer = []$;
2 Append $prev\_memory$ to $replay\_buffer$;
3 **Forward Pass**:;
4 **for** $t = 0, 1, 2, \ldots, T-1$ **do**
5     $m_{t+1} = \text{Model}(x_t, m_t)$ ;         `// No gradient computation`
6     $m_{t+1} = \text{astro\_mem\_retention}(m_{t+1})$ ;       `// Apply astrocytic memory`
      `retention`
7     Append $m_{t+1}$ to $replay\_buffer$;
8 **end**
9 **Backward Pass**:;
10 Initialize $\nabla m_{t+1} = 0$;
11 **for** $t = T, T-1, \ldots, 0$ **do**
12     $m_{t+1}, o_t = \text{Model}(x_t, m_t)$ ;          `// Recompute outputs`
13     Compute $loss = \text{loss\_function}(o_t)$;
14     Perform backpropagation $loss.backward()$;
15     Backpropagate through memory $m_{t+1}.backward(\nabla m_{t+1})$ ;      `// Compute` $\nabla m_t$
16 **end**
17 Save $m_T$ for the next rollout's update;

---

Table S1: Neuroscience Macro-model Hyperparameters

| Parameter | Value |
|---|---|
| **Network Details** | |
| Number of presynaptic neurons | 3 |
| Number of presynaptic neurons | 3 |
| Number of astrocytes | 1 |
| Timescale | $300\ s$ |
| Timestep, $dt$ | $0.04\ s$ |
| **Neural Dynamics** | |
| Neural dynamics timescale , $\tau_n$ | $0.5\ s$ |
| Membrane potential threshold, $V_{th}$ | $1\ mV$ |
| Reset potential, $V_{reset}$ | $-1\ mV$ |
| Decay parameter, $\lambda$ | 0.2 |
| Bias parameter, $b$ | 0 |
| Non-linearity, $\phi$ | $tanh$ |
| **Synaptic Dynamics** | |
| Synaptic dynamics timescale , $\tau_s$ | $0.75\ s$ |
| Decay parameter, $\beta$ | 0.25 |
| Bias parameter, $c$ | 0 |
| Non-linearity, $\theta$ | $tanh$ |
| **Astrocytic STP Dynamics** | |
| Short-term astrocytic process dynamics timescale , $\tau_p^s$ | $1\ s$ |
| Decay parameter, $\gamma^s$ | 0.2 |
| Bias parameter, $d$ | 0 |
| Non-linearity, $\psi$ | $tanh$ |
| **Astrocytic LTP Dynamics** | |
| Long-term astrocytic process dynamics timescale , $\tau_p^l$ | $6\ s$ |
| Decay parameter, $\gamma^l$ | 0.1 |
| Non-linearity, $\kappa$ | $sigmoid$ |

## C  HYPERPARAMETER DETAILS

We develop the neuroscience macro-model and simulate the neural, synaptic and short and long-term astrocytic dynamics to realize the neuroscience-algorithm co-design framework achieved by the RMAAT structure. Table S1 shows the hyperparameters involved in simulating the model. For the RMAAT framework, Table S2 presents the hyperparameters used for training and evaluating the IMDB Dataset, while Table S3 details the hyperparameters for the LRA Text Dataset.

Table S2: IMDB Dataset Hyperparameters

| Parameter | Value |
|---|---|
| **Training Parameters** | |
| Batch size | 64 |
| Maximum sequence length in a segment, $N$ | 256 |
| Number of epochs | 20 |
| Learning rate | $1.5e^{-5}$ |
| **Model Architecture** | |
| Embedding dimension, $d$ | 768 |
| Number of heads | 6 |
| Number of neurons of feed forward network | 1024 |
| Number of encoder layers | 3 |
| Dropout | 0.1 |
| **AMRB Parameters** | |
| Number of segments | 6 |
| Number of memory tokens | 32 |
| **Astromorphic Self-Attention Parameters** | |
| Hidden layer neuron number, $m$ | 100 |
| Nonlinearity, $\alpha$ | 0.25 |

Table S3: LRA Text Dataset Hyperparameters

| Parameter | Value |
|---|---|
| **Training Parameters** | |
| Batch size | 64 |
| Maximum sequence length in a segment, $N$ | 512 |
| Number of epochs | 40 |
| Learning rate | $1.2e^{-5}$ |
| **Model Architecture** | |
| Embedding dimension, $d$ | 256 |
| Number of heads | 2 |
| Number of neurons of feed forward network | 1024 |
| Number of encoder layers | 1 |
| Dropout | 0.1 |
| **AMRB Parameters** | |
| Number of segments | 8 |
| Number of memory tokens | 32 |
| **Astromorphic Self-Attention Parameters** | |
| Hidden layer neuron number, $m$ | 100 |
| Nonlinearity, $\alpha$ | 0.25 |

