# OpenReview forum: "RMAAT: A Bio-Inspired Approach for Efficient Long-Context Sequence Processing in Transformers"
_ICLR.cc/2025/Conference — ICLR 2025 Conference Withdrawn Submission_

### Official Review · Reviewer_vupY · 2024-10-16

**Soundness:** 2
**Presentation:** 2
**Contribution:** 2
**Rating:** 3
**Confidence:** 4

**Summary:**

This paper proposes RMAAT (Recurrent Memory Augmented Astromorphic Transformer), a new transformer architecture inspired by astrocyte cells in the brain. The key ideas are:

- Using "astrocytic memories" as a form of compressed context to handle long sequences
- A bio-inspired compression mechanism based on short-term and long-term plasticity
- An "Astrocytic Memory Replay Backpropagation" (AMRB) algorithm for efficient learning

The authors evaluate RMAAT on sentiment classification (IMDB) and text classification (LRA), showing improvements in accuracy, memory usage, and speed compared to baselines.

**Strengths:**

- Novel bio-inspired approach to improving transformers, drawing interesting connections to neuroscience
- Proposed astrocytic memory mechanism provides an elegant way to compress and propagate context
- AMRB algorithm shows promise for more efficient training of long-context models
- Empirical results demonstrate clear improvements over standard and efficient transformer baselines

**Weaknesses:**

- The core concept bears strong similarities to existing work on Retentive Networks [1], particularly in Equations 6 and 8. The authors should clarify how their approach differs from and improves upon RetNet and other related work like GLA (Gated Linear Attention) [2].

- The origin and computation of the R variable, first appearing in line 305, is unclear. This is a critical component that needs to be explained in detail (correct me if I'm wrong)

- While the authors frame their training approach as biologically-inspired "Hebbian Plasticity", the actual training still relies on backpropagation through time (BPTT). The claimed memory savings of AMRB compared to BPTT are actually a common feature of linear transformers using parallel scan (chunk) training, not a novel contribution.

- For IMDB, the maximum total sequence length of 1536 tokens (256 * 6 segments) is not particularly long by modern standards.
Only the text classification task from LRA is evaluated, despite the paper's focus on "efficient long-context sequence processing". A full evaluation on all LRA tasks would provide a more comprehensive picture.

- The biological framing, while interesting, sometimes feels like packaging of existing linear transformer concepts rather than truly novel bio-inspired algorithms. The authors should more clearly delineate which aspects are novel contributions vs. reframing of known techniques.

In summary, while the paper presents some interesting ideas and promising results, there are some concerns about novelty and thoroughness of evaluation that should be addressed. The authors should clarify the key differences from existing work, provide more details on critical components like the R variable, and conduct more comprehensive experiments on truly long-context tasks.


[1]: Sun, Yutao, et al. "Retentive network: A successor to transformer for large language models." arXiv preprint arXiv:2307.08621 (2023).

[2]: Yang, Songlin, et al. "Gated linear attention transformers with hardware-efficient training." ICML 2024.

**Questions:**

See my weakness part.

---

### Official Review · Reviewer_LDJE · 2024-11-01

**Soundness:** 2
**Presentation:** 3
**Contribution:** 2
**Rating:** 5
**Confidence:** 4

**Summary:**

The paper presents a novel approach to computational modeling by introducing a brain-inspired astromorphic transformer that leverages astrocytic mechanisms, integrating both short-term and long-term plasticity to improve processing efficiency and handle longer sequences in machine learning and NLP tasks. The authors propose the Recurrent Memory Augmented Astromorphic Transformer (RMAAT), which utilizes astrocytic nonlinearity and memory retention. This approach addresses the limitations of existing transformers, which struggle with high memory and computational demands when handling long contexts. Experimental results on the Long Range Arena benchmark and IMDB dataset show that the proposed model achieves significant reductions in memory usage and latency, highlighting the potential of biologically inspired techniques in enhancing computational models' efficiency.

**Strengths:**

* The integration of astrocyte cell mechanisms introduces a novel, neuroscience-inspired approach to addressing computational challenges in deep learning, particularly for long-sequence tasks.

* The proposed model provides an alternative to self-attention by aiming to reduce the quadratic complexity growth that limits traditional transformer architectures, thereby improving efficiency.

* The model effectively combines astrocytic functions with short-term and long-term plasticity mechanisms, enhancing the ability to retain and process long-range dependencies.

* Experimental results indicate that this approach achieves notable improvements in memory utilization and computational latency, making it a potentially valuable contribution to bioplausible AI research.

**Weaknesses:**

* The proposed method lacks cohesion, incorporating multiple bio-inspired mechanisms—short-term and long-term plasticity, Hebbian plasticity, and presynaptic plasticity—without clearly organizing these elements into a unified or systematic framework.

* While the paper claims to integrate brain-inspired mechanisms within a transformer architecture, the resulting model largely resembles a linear recurrent neural network, offering limited differentiation from existing linear models like RWKV and RetNet.

* The experimental validation is limited, with only two benchmark comparisons. This restricts the persuasiveness of the results, leaving questions about the generalizability and practical effectiveness of the proposed method.

**Questions:**

1. In Section 3.2, Figure 2 does not appear to align closely with the content discussed. Could you clarify where Hebbian plasticity, presynaptic plasticity, and spatial mapping are represented in Figure 2? It would be helpful to explicitly indicate where these elements are incorporated into the architecture.

2. Could you provide more clarity on the concept of "memory tokens"? Are these similar to KV caches in traditional transformers? Additionally, what distinct roles do memory tokens and astrocytic memory serve in your model?

3. Given that this work integrates multiple bio-inspired mechanisms, could you provide ablation experiments to demonstrate the individual performance impact of each bio-inspired rule adopted? Such an analysis would help in understanding the contribution of each mechanism to the overall model performance.

---

### Official Review · Reviewer_1s99 · 2024-11-04

**Soundness:** 2
**Presentation:** 2
**Contribution:** 2
**Rating:** 3
**Confidence:** 4

**Summary:**

This paper proposes a novel transformer-based architecture that maps properties of astrocytes, a neural cell that has recently been discovered to play an important role in various central cognitive processes, onto a novel version of the series of models called “Astromorphic Transformers”. One of the central contributions to the architecture seems to be the introduction of a linear-memory relative positional encoding mechanism and the introduction of long-term memory tokens which in some sense mimic the behavior of continuous-time models of neuron-astrocyte dynamics. The architecture is evaluated on two versions IMDB movie reviews dataset in terms of its hardware efficiency and accuracy.

**Strengths:**

- The paper provides a good overview of the continuous-time model of neuron-astrocyte network dynamics. The simulation helps understand the behavior of the coupled differential equations.

- The implementation of the proposed mechanism is more hardware-efficient than traditional transformers, as well as the Linear Transformer (Mia et al., 2023).

- The paper proposes a mechanism for relative positional encoding in linear transformers. I am not sure how well-addressed this issue is in the literature. Solutions have been proposed in 2021 already [1], but the mechanism proposed here seems to be novel, although I do have a question about its novelty in the “Questions” section.

[1]: Liutkus, Cifka et al., 2021, Relative Positional Encoding for Transformers with Linear Complexity

**Weaknesses:**

- Figure 2 is quite messy. There are lines crossing each other, text slightly covered by certain shapes, and two arrows that point from the astrocyte cell outwards which I cannot decipher. The presentation would greatly benefit from a clearer figure.

- Rather than going into the details of the neuroscience and chemistry of the synaptic plasticity and astrocyte dynamics, the text would greatly benefit from a clearer presentation of the previous work on astromorphic transformers.

- The reported fivefold reduction in memory utilization holds with respect to a baseline that can be put in question. It is not entirely clear what the reported “linear transformer” is, and it is very much not clear to me why a linear version of the transformer would consume approximately 3-4x as much memory as the full quadratic softmax transformer does.

- The two benchmarks are essentially one and the same. It is not entirely clear to me what the difference is, as LRA text classification is exactly a dataset of IMDB reviews. It seems that the LRA version is longer. Also, I would not call this a “key benchmark” in any circumstances.

- Because of the limited description of the previous work on astromorphic transformers, it is difficult to judge what in the paper is novel and what is an explanation of previously proposed models.

**Questions:**

- Starting at line 196, the long-term process dynamics are said to be a key contribution that enables the temporal integration of astrocytes within the proposed architecture. I am slightly confused by the use of the term "contribution", as I was under the impression that the long-term process dynamics are established in the literature. Could the authors clarify whether the long-term process dynamics are a contribution of their work, or whether this is an already known model in neuroscience?

- Astrocytic memory is very difficult to understand based on the provided description. Could you provide an equation describing how the memory tokens are initialized, and integrated into the transformer, and how the memory retention factor influences these tokens?

- Is the phi(R) encoding of relative positional information a novel contribution of this work? I am somewhat under the impression that it is not and that the proposed novelty is the mapping of this matrix to certain functions in the continuous-time model described previously.

Limitations:

- While the neural model is explained well, I find that the paper falls short of adequately motivating the mapping from the neural dynamics onto the deep learning model. For example, starting at line 308 of the paper, what prompts the authors to associate Hastro to psi(pijs), and what is the insight contained in the proposed association?

- The experimental evaluation is rather limited and seem to essentially be constrained to an analysis on IMDB movie review sentiment classification. Could the authors clarify what the difference between the two benchmarks they used is? Additionally, the authors may consider evaluating their architecture on the rest of the LRA dataset, as that would provide a clearer picture of the performance of the model. Resources permitting, reporting language modeling perplexity with various context lengths on datasets such as WikiText-103 or EnWik8 would improve the quality of the evaluation. The evaluation is unfortunately very limited in its current state.

---

### Official Review · Reviewer_au8d · 2024-11-08

**Soundness:** 2
**Presentation:** 1
**Contribution:** 2
**Rating:** 3
**Confidence:** 3

**Summary:**

In this work, the authors propose biologically inspired modifications to self-attention based on astrocytic processing, to enable longer context handling. They modify self-attention in a number of ways to achieve an architecture that is then tested on various tasks including some tasks in long-range arena.

**Strengths:**

The mechanisms used and their combination seems novel. The performance of the proposed method is good.

**Weaknesses:**

The primary weakness of the paper is that it doesn't clearly separate out the biological terminology from the model terminology, which makes it very hard to follow. It would have been useful to have a section that summarizes all the changes mathematically, and separately discuss how they are related to astrocytes.

Another weakness is that the empirical evaluation is very limited -- the proposed architecture is not compared with more recent state-space models, and the comparison is not done on all the LRA benchmarks.

**Questions:**

See weaknesses.

---

### Note · Authors · 2024-11-20

I have read and agree with the venue's withdrawal policy on behalf of myself and my co-authors.